# Global Development of Research on Anorectal Malformations over the Last Five Decades: A Bibliometric Analysis

**DOI:** 10.3390/children9020253

**Published:** 2022-02-14

**Authors:** Illya Martynov, Xiaoyan Feng, Johannes W. Duess, Jan-Hendrik Gosemann, Martin Lacher, Steffi Mayer

**Affiliations:** Department of Pediatric Surgery, University of Leipzig, 04103 Leipzig, Germany; xiaoyan.feng@medizin.uni-leipzig.de (X.F.); johannes.duess@uniklinik-leipzig.de (J.W.D.); jan-hendrik.gosemann@medizin.uni-leipzig.de (J.-H.G.); martin.lacher@medizin.uni-leipzig.de (M.L.); steffi.mayer@uniklinik-leipzig.de (S.M.)

**Keywords:** anorectal malformations, ARM, posterior sagittal anorectoplasty, PSARP, bibliometrics

## Abstract

Purpose: Anorectal malformations (ARM) are one of the most challenging congenital malformations in pediatric surgery. We aimed to assess the research activity on ARM over the last five decades. Methods: Data on original research publications were retrieved from the Web of Science Core Collection (1970–2020), and analyzed for countries, authors, scientific journals, and top-ten papers. Scientific quantity was assessed by the number of publications. Research quality was estimated from the number of citations, average citation rate per item (ACI), and h-index. Results: A total number of 1595 articles with 19,419 citations (ACI = 12.2; h-index = 54) were identified. The annual number of publications and citations significantly increased over time (*p* < 0.0001). The USA (*n* = 386; 24.2%), Japan (*n* = 153; 9.6%), and China (*n* = 137; 8.6%) were the most productive countries; and the USA (*n* = 7850; ACI = 20.3; h-index = 44), Japan (*n* = 1937; ACI = 12.6; h-index = 21), and the Netherlands (*n* = 1318; ACI = 17.3; h-index = 22) were the top cited countries. Articles were preferentially published in JPS (*n* = 391; 24.5%), PSI (*n* = 181; 11.3%), and EJPS (*n* = 56; 3.5%). Top-ten cited papers focused on classification (*n* = 1), surgical technique (*n* = 3), associated syndromes (*n* = 2), postoperative outcome (*n* = 3), and basic research (*n* = 1). Conclusion: This bibliometric study provides valuable insights into the global development of ARM research, and shows that clinical studies and international collaborations dominate in this field.

## 1. Introduction

Anorectal malformations (ARM) are congenital anomalies of hindgut development occurring in 2 to 6 per 10,000 live births [1]. Depending on the type of ARM, a variety of different management strategies and surgical procedures are performed. The surgical procedure for low types of malformations is a posterior sagittal anorectoplasty (PSARP), firstly described by Pieter deVries and Alberto Peña in 1982 [2]. However, when the rectourinary fistula is above the level of coccyx in male patients (high rectoprostatic fistula, rectobladderneck fistula), it can only be reached and divided via a transabdominal approach. In these cases, a laparoscopically-assisted endorectal pull-through (LAARP) can be performed [3]. 

ARM patients are at long-term risk for impaired health-related quality of life (HrQoL) in both physical and psychosocial aspects [4,5]. Thus, surgical treatment and care for ARM need to be constantly improved by scientific studies and high-quality publications. Research on ARM also includes topics such as embryology, genetics, surgical techniques, and long-term outcomes to understand the underlying mechanisms of disease development, and to provide new treatment strategies. 

Given the enormous volume and heterogeneity of ARM-related publications, an assessment of the scientific literature on this topic is crucial. In contrast to review articles, which summarize previously published studies and gather the existing knowledge of a field, the discipline of bibliometrics provides the insights into a global research productivity by using mathematical and statistical methods [6]. In general, the quantity of research is assessed by absolute number of publications. The quality of research is determined by the frequency of citations of papers. Furthermore, specific parameters such as the h-index, a common proxy measure for individual scientific output, and impact index have been introduced to specify an individual’s scientific output while considering citation counts and years since publication [7,8]. Additionally, interrelations between bibliometric parameters such as research topics, striking journals, active countries, and prolific authors or collaborations can be illustrated using innovative visualization tools. 

In this bibliometric study, we aimed to evaluate the scientific quality and quantity including the hot topics of original articles related to ARM. We assessed the research activity by numbers of publications and citations, thereby identifying the most productive countries, researchers, and journals that frequently publish articles on ARM. In addition, the top cited papers and the global collaboration networks were analyzed, providing an overview on research activities in the context of ARM. 

## 2. Materials and Methods

Original peer-reviewed scientific publications on ARM between 1970 and 2020 were identified using the Web of Science Core Collection™ (www.webofknowledge.com, Clarivate Analytics, Boston, MA, USA) by two independent reviewers (IM, XF) on 18 July 2020, updated on 19 November 2020. To determine relevant research items only, a “title” rather than “topic” search approach was used: “anorectal malformation(s)” OR “congenital anorectal malformation(s)” OR “congenital anorectal abnormality” OR “anorectal atresia” OR “imperforate anus” OR “cloacal malformation(s)” OR “cloaca” OR “posterior sagittal anorectoplasty” OR “PSARP” OR “anoplasty” OR “anorectoplasty” OR “anterior sagittal anorectoplasty” OR “laparoscopic-assisted posterior sagittal anorectoplasty” OR “laparoscopic assisted posterior sagittal anorectoplasty” OR “LAARP” OR “laparoscopic assisted PSARP” OR “laparoscopic-assisted PSARP” OR “laparoscopic-assisted anorectoplasty” OR “laparoscopic assisted anorectoplasty”, NOT “colonic atresia”, NOT “small bowel atresia”, NOT “duodenal atresia” [9]. To assess the ongoing research activities on ARM, only original articles were included. Proceeding papers, editorial materials, meeting abstracts, letters, and review articles were excluded from the analysis. 

From the generated data, the following variables were extracted: countries, authors, journals, and year of publication. Furthermore, research quantity and quality were evaluated. Research quantity was defined as the absolute number of publications per researcher and country as described by Zamarripa et al. [10]. Research quality was defined as the absolute number of citations, the average number of citations per item (ACI, i.e., the number of citations divided by the number of publications), impact factor (IF), the h-index related to ARM research, and the impact index. The h-index, a common proxy to measure an individual scientific output, is defined as the number of papers (h) published by one researcher that have been cited at least h-times. The h-index can also be applied for groups of scientists, generating an institution-, journal-, or country-level metric that accounts for both the quantity (i.e., the number of papers) and quality (i.e., the citations of these papers) of publications [7,11]. A higher h-index indicates a higher scientific consideration. The impact index is obtained by dividing the number of years since publication by the number of citations, multiplied by 100 [8]. A lower impact index expresses a higher scientific power. 

Visual bibliometric mapping and citation analyses were performed with VOSviewer version 1.6.5 (www.vosviewer.com). The size of the frame in the VOSviewer diagram represents the number of publications per country, the colors indicate clusters, and the lines represent collaborations. The link strength between the frames refers to the frequency of co-occurrence. The total link strength is the sum of link strengths per country (frame). The international collaboration network mapping was generated from all countries that had at least 25 publications with total link strength of more than 10. The Spearman correlation coefficient was used to test correlations between selected continuous variables. Statistical analyses were performed with SPSS v. 23 (SPSS 23.0—SPSS Inc., Chicago, IL, USA) and GraphPad Prism v. 6.01 (GraphPad, La Jolla, CA, USA). *p* values of <0.05 were considered statistically significant. 

## 3. Results

### 3.1. Total Output 

We identified an overall number of 1595 original articles and 19,419 citations published between 1970 and 2020 on ARM. Among these articles, the average number of citations per item was 12.2, with an overall h-index for ARM research of 54. That means, of all papers published on ARM, there are at least 54 articles that have been cited at least 54 times. There was a continuous increase in the number of annual publications from 1970 (*n* = 11) to 2020 (*n* = 81) (r^2^ = 0.53, *p* < 0.0001), and citations from 1971 (*n* = 2) to 2020 (*n* = 1456) (r^2^ = 0.79, *p* < 0.0001) (Figure 1).

Overall publication and citation performance of ARM research from 1970 to 2020, presented as the total number of publications (orange) and citations (red) per year. Landmarks in the treatment of ARM are indicated. PSARP: posterior sagittal anorectoplasty, LAARP: laparoscopically-assisted endorectal pull-through. 

### 3.2. Top Productive Countries and Authors

The ten most productive countries as assessed by the absolute number of publications were the USA (*n* = 386; 24.2%), Japan (*n* = 153; 9.6%), China (*n* = 137; 8.6%), India (*n* = 93; 5.8%), Italy (*n* = 93; 5.8%), England (*n* = 85; 5.3%), the Netherlands (*n* = 76; 4.8%), Germany (*n* = 67; 4.2%), France (*n* = 57; 3.6%), and Canada (*n* = 47; 2.9%). Publications from the USA received the highest number of citations (*n* = 7850; ACI = 20.3; h-index = 44), followed by Japan (*n* = 1937; ACI = 12.6; h-index = 21), and the Netherlands (*n* = 1318; ACI = 17.3; h-index = 22) (Table 1). The most productive authors of original research articles on ARM were Alberto Peña (*n* = 64, h-index = 29), Marc Anthony Levitt (*n* = 53, h-index = 20), and Wei Lin Wang (*n* = 42, h-index = 10) (Table 2). Four of the most productive authors were from China, three from the USA, two from European countries, and one from Japan. 

### 3.3. Top Cited Journals and Articles 

A total of 100 journals have published original research articles on ARM; the most productive journals with a mean IF of 2.2 are summarized in Table 3. Five of the top ten journals were related to Pediatric Surgery, namely the *Journal of Pediatric Surgery* (*n* = 391; 24.5%; IF = 2.545 (2021)), *Pediatric Surgery International* (*n* = 181; 11.3%; IF = 1.827 (2020)), the *European Journal of Pediatric Surgery* (*n* = 56; 3.5%; IF = 2.191 (2021)), the *Journal of Laparoendoscopic and Advanced Surgical Techniques* (*n* = 20; 1.2%; IF = 1.878 (2020)), and *Seminars in Pediatric Surgery* (*n* = 17; 1.1%; IF = 2.807 (2020)). 

The ten most cited original research articles on ARM comprised the Krickenbeck classification system of ARM phenotypes (*n* = 1), the first description and first reevaluation analysis of PSARP (*n* = 2), the description of syndromes associated with ARM (*n* = 2), the clinical outcome after ARM surgery (*n* = 3), the introduction of LAARP (*n* = 1), as well as one basic research paper on the genetic background of ARM in mice (*n* = 1). All top cited articles were published between 1972 and 2005, and nine of them were clinical studies/case studies. The individual impact index, which adjusts for the number of citations over time since publication, correlated with the number of citations, indicating the impact of the particular scientific work irrespective of the period of availability (Table 4). 

### 3.4. Cooperation among Countries and Authors

Three different clusters of international collaborations could be identified. Cluster 1 comprised the USA, China, Japan, Canada, and Australia; Cluster 2 included England, France, Germany, Italy, the Netherlands, and Switzerland; whereas Cluster 3 involved Austria, Israel, Spain, and Sweden (Figure 2).

Collaboration network of ARM researchers. There are three clusters of collaboration countries: green indicates Cluster 1 (the USA, China, Japan, Canada, Australia), red indicates Cluster 2 (England, France, Germany, Italy, the Netherlands, Switzerland), and blue indicates Cluster 3 (Austria, Israel, Spain, Sweden). Line thickness represent the strength of collaboration between countries, and frame size indicates the total number of publications per country.

## 4. Discussion

We conducted the first bibliometric analysis on original research activities on ARM between 1970 and 2020, and identified 1595 articles with 19,419 citations, mostly originating from the USA, China, and Japan, that were often published in Pediatric Surgery related journals. Nine of the ten most cited articles focused on clinical aspects (classification, surgical technique, associated syndromes, and outcome), and only one published basic research results from an animal model. 

In line with bibliometric analyses of other topics in pediatric surgery such as esophageal atresia (2170 publications; 26,755 citations; 1945 to 2018; peer-reviewed articles) and congenital diaphragmatic hernia (CDH; 3669 publications; citation N/A; 1910 to 2016; peer-reviewed articles), we also found a significant increase in the number of publications and citations during the last five decades [9,20]. This excessive increase in publications may be explained by a growing interest of clinicians and scientists to share their findings within the scientific community. Moreover, the quality and quantity of publications are a marker of individual academic success; people who want to become professional in their field have strong ambitions to be involved in research activities that are rewarded by publication [21].

The USA, China, and Japan were the three most active countries publishing on ARM. The USA had the largest output of publications, which may be promoted by the allocation of high budgets for research with a vast number of research centers [22]. Correspondingly, in several bibliometric studies on various medical topics, the USA ranked first in research productivity [20,23,24]. China was the second most productive country on ARM research. Chinese studies reported frequently on basic ARM research including genetic pathways in animal models or the evolution of single-incision laparoscopic approaches to ARM [25,26,27,28,29,30,31,32]. The high number of Japanese papers may be explained by the establishment of the Japanese Group for the Study of Anorectal Anomalies (JGSA) in 1974. The JGSA group was found to systematically assess ARM with regards to standardized surgical techniques and indications for clinical as well as radiological investigations [33]. This group is comparable to the European ARM-Net consortium, a group of professionals and patient representatives that syndicated in 2010 for the exchange of data and knowledge to improve clinical care, as well as research on ARM [34].

Authors from the USA maintained a high number of collaborations for ARM research with China, Japan, Canada, and Australia. As a result, these countries were most productive in the number of publications and citations except for Canada. Accordingly, international collaborations among scientists are gaining importance to advance patient care [35,36]. Especially for ARM, which is a rare congenital disorder with a wide spectrum of subtypes, international collaborations are crucial for conducting well-powered clinical trials and preclinical translational studies [37].

The most productive authors on ARM research with the highest citation and h-indices were Alberto Peña, Marc Anthony Levitt, and Wei Lin Wang. Alberto Peña and Pieter de Vries first described the PSARP procedure in 1982 [2]. Marc Anthony Levitt, who is the second top productive author according to the number of publications, and Alberto Peña have been coworkers for years. Their main scientific interests have been surgical techniques, as well as clinical studies on the clinical outcome after correction of ARM [38,39,40,41]. In contrast, Wei Lin Wang mainly published on basic research, i.e., embryonic development of ARM [42,43,44,45]. 

However, the identification of “relevant” authors by the absolute number of publications and citations, as well as by h-index, is problematic. Authors who may publish only a few papers but in high-impact journals with a high number of corresponding citations will still have a significantly lower h-index compared to authors who received the same number of citations deriving from more papers published in moderate-impact journals [46]. Although the h-index remains an important bibliometric indicator to assess an individual output, its parameters should be taken into account when it is applied to assess the scientific “importance” of a researcher. This includes its size dependency (i.e., the total number of publications and citations), the lack of field-normalization (i.e., researchers active in different scientific fields), and its reliance on databases (i.e., Google Scholar h-index, Scopus h-index, Web of Science h-index) [47,48,49]. 

Nine of the ten most cited articles focused on clinical aspects, including classification of ARM phenotypes (*n* = 1), surgical technique (*n* = 3), outcome following ARM repair (*n* = 3), syndromes associated with ARM (*n* = 2), and basic science (*n* = 1). 

### 4.1. Classification of ARM Phenotypes 

The most cited manuscript, with 298 citations, was “Preliminary report on the international conference for the development of standards for the treatment of anorectal malformations” by Holschneider A, Hutson J, Peña A et al., which was published in 2005 in the Journal of Pediatric Surgery [12]. In this paper, the authors reported on an expert meeting in Krickenbeck, Germany, where a uniform international classification system for ARM (i.e., Krickenbeck classification) was established. This classification is still used today, and paved the way to a uniform scoring system for structured treatment and comparable follow-ups for ARM patients. 

### 4.2. Surgical Technique 

Three of the ten most-cited papers (759 of 1922 citations (39.5%)) reported on novel surgical techniques for the repair of ARM. In October 1982, Pieter deVries and Alberto Peña first described PSARP for the repair of high ARMs, which was successfully performed 34 patients [2]. Interestingly, two months later, in December 1982, Alberto Peña and Pieter deVries published the second papers discussing many technical details of the PSARP procedure [13]. Therefore, the initial descriptions of PSARP are by far the most cited topic in ARM research. Also, the initial report on the laparoscopically-assisted anorectal pull through technique in 11 children with high imperforate anus by Keith Georgeson et al. in 2000 ranks among the ten most cited publications [3]. This paper has been cited 179 times since then. It paved the way to minimal invasive procedures in ARM. Better visualization and dissection of the fistula, rectal mobilization and resection, accurate tunnel formation, as well as fewer wound infections may be advantages of LAARP [50]. In 2017, a meta-analysis by Han et al. compared the outcomes of 191 LAARP and 169 PSARP procedures in high and intermediate ARM, which showed shorter hospital stay, and less postoperative complications and wound infections for LAARP. However, the differences in operative time, rectal prolapse, anal stenosis, and anorectal manometry were not conclusive [51]. To date, long-term follow up studies on the impact of LAARP on bowel function quality of life are missing.

### 4.3. Outcome following ARM Repair 

Three of the top-ten citations focused on the long-term outcome of ARM patients with regards to bowel function [16,17,19]. The authors stated that normal bowel function can be expected in up to 75% of patients with ARM. Even in a significant proportion of children with high or intermediate ARM, an adequate bowel function is possible. These citations reflect the importance of, and attention to, the postoperative care in children with ARM, thus contributing to the improvement of treatment standards for these patients. 

### 4.4. Syndromes Associated with ARM 

Two of the ten most cited papers reported on syndromes associated with ARM. Philip Townes and Eric Brocks firstly described the “Hereditary syndrome of imperforate anus with hand, foot, and ear anomalies”, which later became known as Townes–Brocks Syndrome [52]. Judith Hall and Philip Pallister described six sporadic cases of patients with ARM and associated malformations, including congenital hypothalamic hamartoblastoma, hypopituitarismus, and postaxial polydactyly, in 1980 [14]. Since then, this combination of symptoms has become known as the Pallister–Hall syndrome [53]. These often-cited studies underline that clinicians should always be are aware of syndromic ARM cases with associated anomalies. 

### 4.5. Basic Science

Only one of the ten most cited papers reported on basic science, which was published in the American Journal of Pathology (IF = 3.5) in 2001 with a total number of 163 citations [15]. Here, the authors demonstrated that Sonic hedgehog (Shh) signaling is essential for normal development of the distal hindgut in mice. Accordingly, mutations affecting Shh signaling result in a spectrum of anorectal malformations. These findings were later confirmed by Zhang et al. in children with ARM, suggesting that down-regulation of the Shh signaling may be related to the occurrence of high-type ARM [54].

Several aspects may explain the relevant difference in the number of top-citations between clinical and basic research publications. Studying the underlying embryology and molecular biology in humans with ARM is difficult due to the scarcity of tissue materials. Consequently, one may ask for an appropriate animal model that mimics the disease well, as is available for other congenital malformations such as the nitrofen-rat model for CDH [55]. For ARM, there are also different animal models available: Danforth et al. described a mouse model (so-called “Danforth’s short-tail mouse”) caused by a spontaneous mutation in chromosome 2 [56]. ARM-like lesions can also be induced in pregnant rodents by maternal administration of ethylenethiourea (ETU), etretinate, or Adriamycin [57]. Different treatment options, such as the impact of folic acid supply on the incidence of ARM in ETU-treated Wistar rats, have been studied [58]. However, these animal models have low translational value, as they do not address the important issues of ARM patients, which are fecal continence and sexual dysfunction [57]. 

There are some limitations of our study. Firstly, we only used the Web of Science™ data base to search for publications, and thus, other sources may have led to a different number of research items or citation counts. Secondly, due to constant changing citation volumes over time, the results of our study are of temporary nature, and valid for the time point of the data extraction date (18 July 2020) and data update (November 19, 2020) only. Third, the share of non-cited papers should also be considered when determining the h-index. Finally, clinical research from an international perspective would constitute patients from different countries in a collaborative dataset, which is rarely the case in ARM studies to date. International collaborations as depicted by bibliometric analysis can only be suggested from the authors’ institution as listed in the publications. However, global authors contributing to the same manuscript suggest an international interchange. Moreover, bibliometric studies cannot identify the critical or unmet needs of a research topic from the literature. Nevertheless, we believe that our study provides a detailed bibliometric analysis of research activities on ARM. 

## 5. Conclusions

The bibliometric analysis of original research articles on ARM is important for clinicians and scientists to understand global research activities on this topic. Although over the past five decades, clinical ARM research has increased tremendously, basic science studies remained uncommon. Hence, national and international efforts should be made to strengthen fundamental and translational research on ARM.

## Figures and Tables

**Figure 1 children-09-00253-f001:**
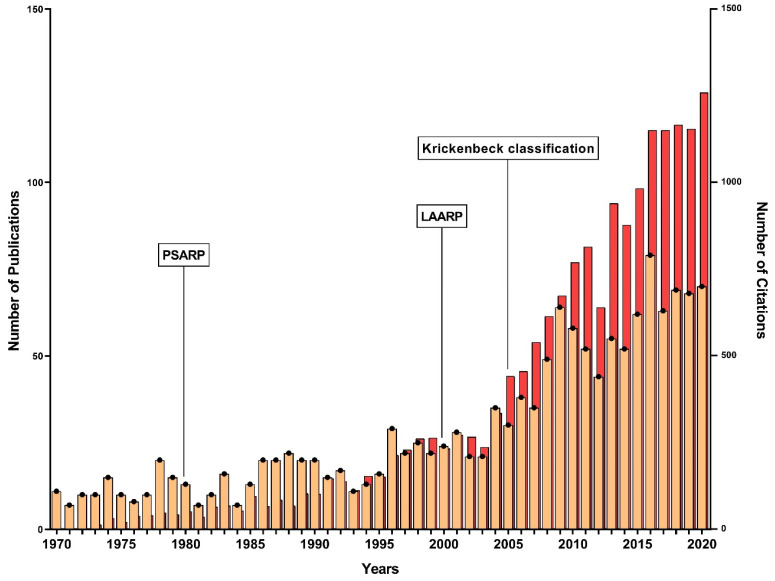
Absolute number of publications and citations on original ARM research during the last 50 years. Abbreviations: PSARP = posterior sagittal anorectoplasty; LAARP = laparoscopically-assisted endorectal pull-through.

**Figure 2 children-09-00253-f002:**
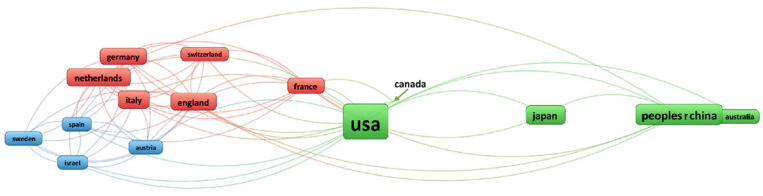
International collaboration network among countries publishing on ARM between 1970 and 2020.

**Table 1 children-09-00253-t001:** Most productive countries publishing on ARM between 1970 and 2020.

	Country	Total Publications *N*	Total Citations *N*	ACI	h-Index *
1	USA	386	7580	20.3	44
2	Japan	153	1937	12.7	21
3	China	137	1094	8.0	18
4	India	93	812	8.7	15
5	Italy	93	1072	11.5	15
6	England	85	1269	15.0	20
7	Netherlands	76	1318	17.3	22
8	Germany	67	1140	17.0	18
9	France	57	365	6.4	11
10	Canada	47	1011	22.0	16

* for publications on ARM only; ACI: average citation rate per item.

**Table 2 children-09-00253-t002:** Most productive authors publishing on ARM between 1970 and 2020.

	Author	Country	Total Publications *N*	Total Citations *N*	h-Index *
1	Peña Alberto	USA	64	2738	29
2	Levitt Marc Anthony	USA	53	1098	20
3	Wang Wei Lin	China	42	370	10
4	Bai Yu Zuo	China	36	276	8
5	Yuan Zheng Wei	China	29	357	10
6	Bischoff Andrea	USA	26	280	11
7	de Blaauw Ivo	Netherlands	25	319	11
8	Naomi Iwai	Japan	25	585	12
9	Rintala Risto	Finland	24	706	15
10	Jia Huimin	China	23	111	7

* for publications on ARM only.

**Table 3 children-09-00253-t003:** Most productive journals reporting on ARM between 1970 and 2020.

	Journal	IF 2019	Total Publications *N*	Total Citations *N*	h-Index *
1	*Journal of Pediatric Surgery*	1.9	391	8106	45
2	*Pediatric Surgery International*	1.6	181	1613	20
3	*European Journal of Pediatric Surgery*	1.7	56	323	9
4	*Journal of Urology*	4.8	43	614	15
5	*Disease of the Colon and Rectum*	3.7	29	415	12
6	*Pediatric Radiology*	1.8	29	381	12
7	*Journal of Laparoendoscopic and* *Advanced Surgical Techniques*	1.4	20	131	7
8	*Seminars in Pediatric Surgery*	2.4	17	352	9
9	*Journal of Ultrasound in Medicine*	0.5	15	177	7
10	*Urology*	1.8	15	88	5

* for publications on ARM only; mean impact factor (IF) = 2.2.

**Table 4 children-09-00253-t004:** Most cited original research publications on ARM between 1970 and 2020.

	Article	Research Field	Author	Year	Journal	Total Citations (*N*)	Impact Index *
1	Preliminary report on the international conference for the development of standards for the treatment of anorectal malformations [12]	Classification of phenotypes	Holschneider A, Hutson J, Peña A et al.	2005	*J Pediatr Surg*	298	4.3
2	Posterior sagittal anorectoplasty [2]	Surgical technique	deVries PA, Peña A	1982	*J Pediatr Surg*	290	5.4
3	Posterior sagittal anorectoplasty—Important technical considerations and new applications [13]	Surgical technique	Peña A, deVries PA	1982	*J Pediatr Surg*	290	5.4
4	Congenital hypothalamic hamartoblastoma, hypopituitarism, imperforate anus, and postaxial polydactyly—A new syndrome? [14]	Syndromic ARM	Hall JG, Pallister PD, Clarren SK et al.	1980	*Am J Med Genet*	236	15.6
5	Laparoscopically assisted anorectal pull-through for high imperforate anus—A new technique [3]	Surgical technique	Georgeson KE, Inge TH, Albanese CT	2000	*J Pediatr Surg*	179	9.2
6	Anorectal malformations caused by defects in sonic hedgehog signaling [15]	Basic science	Mo R, Kim JH, Zhang J et al.	2001	*Am J Pathol*	163	10.1
7	Is normal bowel function possible after repair of intermediate and high anorectal malformations? [16]	Outcome (bowel function)	Rintala RJ, Lindahl H	1995	*J Pediatr Surg*	132	11.5
8	Advances in the management of anorectal malformations [17]	Outcome (bowel function)	Peña A, Hong A	2000	*Am J Surg*	131	17.3
9	Hereditary syndrome of imperforate anus with hand, foot, and ear anomalies [18]	Syndromic ARM	Townes PL, Brocks ER	1972	*J Pediatr*	110	32.2
10	Bowel management for fecal incontinence in patients with anorectal malformations [19]	Outcome (bowel function)	Peña A, Guardino K, Tovilla JM et al.	1998	*J Pediatr Surg*	93	36.2

* a lower impact index expresses a higher scientific power.

## Data Availability

Not applicable.

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
