# Peer review of "Global Development of Research on Anorectal Malformations over the Last Five Decades: A Bibliometric Analysis"

_children, 2022, doi:10.3390/children9020253_

Round 1
Reviewer 1 Report
This is an interesting overview research on ARM bibliography.
I feel though that some important papers have been missed- for instance I couldn't find in the basic science section any mention to the following important paper: Embriology of Anorectal Malformations (Kluth D. , Semin Pediatr Surg. 2010).
I overall missed the final end point of this article and I think the authors could better clarify it.
Author Response
We would like to thank the reviewer#1 for the comment. In this bibliometric study only original peer-reviewed scientific publications on ARM were included in the analysis. Consequently, proceeding papers, editorial materials, meeting abstracts, letters, and review articles were excluded from the analysis. Since the paper written by our colleague Dr. Kluth is a review article, it was not included in the analysis. Finally, bibliometric analysis enables researchers to measure, assess and evaluate the publication output and scholarly performance. Due to the descriptive study design of bibliometric analyses, we agree that there is no clear “final end point”. However, we aimed at precisely describing what has been published on ARM during the last 5 decades thus, elucidating the scientific activities on ARM.
Reviewer 2 Report
Dear Sirs
This an interesting view of the literature concerning anorectal malformations in children. The authors were able to identify trends in the number of publications over the years as well as preferred authors, journals and countries that publish on the subject. Because publication of a manuscript is a complex subject, that depends on a lot of factors, more studies like this should be undertaken, that would help to identify if the trends and preferences described reflect only the quality of the manuscripts or if there are any bias that may have influenced the overall distribution of publications in this (or any other) subject.
The only remark I won make is that, in lines 8 and 175 of the manuscript proof the authors used "....fifty decades..." instead of "...five decades...."(or fifty years).
I would like to congratulate the authors for their initiative and for the excellent work.
Author Response
The authors would like to thank the reviewer#2 for carefully reading our manuscript and positive feedback. We have corrected “fifty decades” to “five decades”.